# The Gelatinase Inhibitor ACT-03 Reduces Gliosis in the Rapid Kindling Rat Model of Epilepsy, and Attenuates Inflammation and Loss of Barrier Integrity In Vitro

**DOI:** 10.3390/biomedicines10092117

**Published:** 2022-08-29

**Authors:** Diede W. M. Broekaart, Till S. Zimmer, Sophie T. Cohen, Rianne Tessers, Jasper J. Anink, Helga E. de Vries, Jan A. Gorter, Roger Prades, Eleonora Aronica, Erwin A. van Vliet

**Affiliations:** 1Amsterdam UMC, Location University of Amsterdam, Department of (Neuro)Pathology, Amsterdam Neuroscience, Meibergdreef 9, 1105 AZ Amsterdam, The Netherlands; 2Department of Molecular Cell Biology and Immunology, Amsterdam UMC, Vrije Universiteit Amsterdam, Amsterdam Neuroscience, 1081 HV Amsterdam, The Netherlands; 3Swammerdam Institute for Life Sciences Center for Neuroscience, University of Amsterdam, 1098 XH Amsterdam, The Netherlands; 4Accure Therapeutics S.L., 08028 Barcelona, Spain; 5Stichting Epilepsie Instellingen Nederland (SEIN), 2103 SW Heemstede, The Netherlands

**Keywords:** matrix metalloproteinases, pro-inflammatory factors, brain inflammation, blood–brain barrier, extracellular matrix, astrocytes, microglia

## Abstract

Matrix metalloproteinases (MMPs) are endopeptidases responsible for the cleavage of intra- and extracellular proteins. Several brain MMPs have been implicated in neurological disorders including epilepsy. We recently showed that the novel gelatinase inhibitor ACT-03 has disease-modifying effects in models of epilepsy. Here, we studied its effects on neuroinflammation and blood–brain barrier (BBB) integrity. Using the rapid kindling rat model of epilepsy, we examined whether ACT-03 affected astro- and microgliosis in the brain using immunohistochemistry. Cellular and molecular alterations were further studied in vitro using human fetal astrocyte and brain endothelial cell (hCMEC/D3) cultures, with a focus on neuroinflammatory markers as well as on barrier permeability using an endothelial and astrocyte co-culture model. We observed less astro- and microgliosis in the brains of kindled animals treated with ACT-03 compared to control vehicle-treated animals. In vitro, ACT-03 treatment attenuated stimulation-induced mRNA expression of several pro-inflammatory factors in human fetal astrocytes and brain endothelial cells, as well as a loss of barrier integrity in endothelial and astrocyte co-cultures. Since ACT-03 has disease-modifying effects in epilepsy models, possibly via limiting gliosis, inflammation, and barrier integrity loss, it is of interest to further evaluate its effects in a clinical trial.

## 1. Introduction

Matrix metalloproteinases (MMPs) are extracellular zinc-dependent endopeptidases that rely on calcium for activation [1]. Belonging to a family of about 25 members, several MMPs have been demonstrated to be present in the mammalian brain, of which the so-called gelatinases MMP2 and MMP9 are best studied. MMPs are synthesized as a pro-peptide after which they are secreted into the extracellular matrix (ECM) or remain membrane bound [2]. Cleavage of the pro-peptide, which covers the enzyme’s active site, is required for activation and is often executed by plasminogen activators, furin, or other MMPs [3,4]. When fully activated, MMPs can target a plethora of both extra- and intracellular proteins for cleavage. As their name implies, around one quarter of the known targets include proteins of the ECM [5]. Because of their wide range of targets, MMPs are crucial for the maintenance of ECM homeostasis [2,6,7,8] and are involved in the regulation of several biological processes during normal physiological conditions, such as cell migration [9,10] and synaptic plasticity [11]. However, deregulation of MMPs has been associated with ECM alterations and pathological processes such as neuroinflammation and loss of blood–brain barrier (BBB) function [9,12,13]. Indeed, the remaining three quarters of MMPs’ targets include non-ECM substrates, among which are a multitude of proteins involved in inflammation and immune signaling including chemokines, cytokines, and growth factors [5]. MMPs can cause the activation of these signaling molecules by cleavage of the latent protein form and induce a pro-inflammatory milieu [5,12]. At the BBB, MMPs are known to target not only proteins of the specialized ECM that make up the basement membrane surrounding the vasculature, but also the tight junction proteins that firmly connect endothelial cells and control the barrier’s permeability [13,14,15]. It is therefore no surprise that MMPs have been shown to be involved in many neurological disorders. Increased expression and activity of MMPs are associated with epilepsy, Alzheimer’s disease, multiple sclerosis, and others [2,16,17,18,19,20,21].

Several attempts have been made to modulate the activity of MMPs pharmacologically in order to alleviate their excessive activity in disease [22]. MMP inhibitors were initially tested in the oncology field, where increased MMP activity is associated with cancer progression and metastasis [23]. The first generation of MMP inhibitors were hydroxamate-based broad-spectrum inhibitors, targeting several different MMP family members. Due to the high metabolic liability, off-target metallo-endopeptidase inhibition and the consequent side effects, all the tested MMP inhibitors in clinical trials for cancer treatment have failed up to date [22]. Later generations of inhibitors focused on metabolically stronger, non-hydroxamate or alternative site targeting with varying effects in modulating disease progression [24,25,26]. Recently, we have shown that the novel gelatinase inhibitor IPR-179 [27] (now known as ACT-03) reduces seizure severity, seizure frequency, and seizure duration in two rodent models of temporal lobe epilepsy [28]. This MMP2 and MMP9 inhibitor recognizes the zinc-containing active site of the enzyme using chelating properties, and in the nanomolar range does not interact with any other metalloproteinases. Both in vitro and in vivo studies have shown its stability, barrier permeability, and dose-dependent effect in rodents [27]. Furthermore, ACT-03 treatment attenuated seizure-induced cognitive decline without causing any notable side effects [28].

Since, in our previous study [28], we investigated the effects of ACT-03 on seizure activity, cognition, and neuronal loss, and confirmed that MMP2 and MMP9 were inhibited using activity assays and Western blot analysis, in the present study we were interested in determining the drug’s mechanism of actions that could lead to the reduction of symptoms. Furthermore, by identifying the pathological processes that are modulated by ACT-03 treatment, this study might provide more insight in the potential uses of ACT-03 as a therapeutic target. For this purpose, we studied the effects of ACT-03 treatment on astrogliosis and microgliosis in the brains of kindled rats from our earlier study, and further investigated the cellular and molecular alterations in vitro using astrocyte and brain endothelial cell cultures, with a focus on neuroinflammatory markers and endothelial barrier permeability.

## 2. Materials and Methods

### 2.1. Animals

For this study, the brains of adult male Sprague Dawley rats (Harlan Netherlands, Horst, The Netherlands) were used. These were randomly assigned to different treatment groups: non-kindled controls (*n* = 9), vehicle-treated kindled animals (*n* = 10), and ACT-03-treated kindled animals (*n* = 10), as previously reported [28]. A detailed description of electrode implantation and the kindling protocol can be found in the Appendix A, and in our previous paper describing the effects of ACT-03 on epileptogenesis [28]. Four animals (two vehicle-treated and two ACT-03-treated) were excluded from the study due to poor EEG signal or a damaged headset.

#### 2.1.1. ACT-03 Treatment

Kindled rats were treated daily intraperitoneally (i.p.) with either vehicle (5.0% Tween^®^ 80 in 0.9% saline) or 6 mg/kg ACT-03 (Accure Therapeutics, Barcelona, Spain) for one week [28]. Injections (volume of 10 mL/kg) took place one hour before kindling stimulations on day 1, 2, and 3 between 8:00 and 9:00 A.M (see Figure 1). ACT-03 dosage was selected based on lower behavioral seizure severity in pentylenetetrazole-treated mice when 6 mg/kg and 18 mg/kg of ACT-03 was administered, data not shown. Based on effective inhibition of MMP9 activation with 6 mg/kg of ACT-03, we selected this dose for the rapid kindling rat model.

#### 2.1.2. Immunohistochemistry

Rats were decapitated one day after the re-kindling session (23 days after electrode implantation; 9 days after the last ACT-03 injection). The brain was dissected, and the contralateral hemisphere was stored in 4% paraformaldehyde (PFA) for 7 days. After the hemisphere was paraffin-embedded, sagittal sections of a thickness of 5 µm were cut, including the regions of interest (3.75–3.90 mm from Bregma at a lateral level). Sections were mounted on pre-coated glass slides (Star Frost, Waldemar Knittel, Braunschweig, Germany) and were deparaffinated by three consecutive washes with xylene followed by three washes in ethanol (100%, 96%, 70%). This was followed by a 20 min incubation in 0.03% hydrogen peroxide diluted in methanol to block endogenous peroxidase activity. Slides were washed and transferred to 10 mM sodium citrate, pH 6.0, for a heat-induced epitope retrieval using a pressure cooker at 121 °C for 10 min. After slides were cooled, they were washed three times with phosphate-buffered saline (PBS; pH 7.4). Overnight incubation was performed at 4°C with the following primary antibodies: anti-Vimentin (monoclonal mouse IgG, 1:500, Clone V9, M0725, DAKO/Agilent Technologies Netherlands, Amstelveen, The Netherlands), anti-Iba1 (polyclonal rabbit, 1:2000, 019-19741, WAKO, Richmond, VA, USA), anti-GFAP (polyclonal rabbit, 1:4000, Z0334 DAKO/Agilent Technologies Netherlands, Amstelveen, the Netherlands), anti-MMP9 (polycloncal rabbit, 1:100, MAB13458, Merck Millipore, Burlington, MA, USA), or anti-albumin (polyclonal rabbit, 1:80,000, F0117, DAKO/Agilent Technologies Netherlands, Amstelveen, the Netherlands). After this incubation, sections were carefully washed in PBS. A polymer-based peroxidase immunohistochemistry detection kit (Brightvision plus kit, ImmunoLogic, Duiven, the Netherlands) was used, and sections were incubated in 3,3′-diaminobenzidine substrate solution (1:10 in 0.05 M Tris-HCl, pH 7.6; ImmunoLogic, Duiven, the Netherlands) for 5–8 min to visualize the staining, after which all sections were washed in distilled water. Sections were counterstained with hematoxylin for 5 min and washed with tap water for 4 min before being dehydrated in alcohol and xylene and coverslipped.

For analysis of vimentin stainings, three representative images were taken of each sample from the following four regions of interest (ROI): the CA1 and the dentate gyrus (DG) from the dorsal and well as from the ventral part of the hippocampus. Using ImageJ, the percentage of vimentin-positive area was determined per image, and the average vimentin-positive of each ROI was calculated. For analysis of Iba1 stainings, Iba1-positive cells were counted using a 10 × 10 ocular grid and a light microscope (Olympus BX41TF) at a magnification of 200×. In all ROIs, 64 equivalent square fields (8 × 8 rows) were used for counting. Thereafter, the density of Iba1-positive cells was calculated by dividing the number of cells by the area defined by the grid and expressed as number of cells/0.10 mm². Analysis of GFAP and MMP9 stainings is described in the Appendix A.

### 2.2. Cell Culture

#### 2.2.1. Human Fetal Astrocytes

Human fetal brain tissue aged 14–20 weeks of gestation, were used to derive primary fetal astrocyte-enriched cell cultures. All material was collected from medically induced abortions. Donors concluded a written informed consent for the use of the material for research purposes by the Bloemenhove clinic (Heemstede, the Netherlands) in accordance with the Declaration of Helsinki and the Amsterdam UMC Research Code according to the Medical Ethics Committee. Tissue samples were collected in filtered incubation medium containing DMEM/HAM F10 (1:1; Gibco/ThermoFisher Scientific, Waltham, MA, USA), supplemented with 1% penicillin/streptomycin and 10% fetal calf serum (FCS). Primary cell cultures of astrocytes were prepared as previously described [29,30]. Briefly, meninges and large blood vessels that were macroscopically visible were removed before material was mechanically minced into smaller pieces. Material was collected in a tube and enzymatically digested using 2.5% trypsin (Sigma-Aldrich, St. Louis, MO, USA) in incubation medium for 30 min at 37 °C. Material was washed twice using incubation medium and triturated by passing through a 70 µm mesh filter. Cells were collected in culture flask and incubated at 37 °C, 5% CO_2_ for 48 h to let glial cells adhere before washing thoroughly with PBS to remove excess myelin and cell debris. Biweekly refreshment of the culture medium ensured proper cell proliferation. At confluence of 80–90%, cultures were passaged. In the present study, passages 2 to 5 were used. Previous studies have shown that over 98% of the cells in primary culture were immunoreactive for several astroglial markers including glial fibrillary acid protein (GFAP; DAKO Z0334) and S100β (Sigma, 070M4767) [31]. Throughout this study, we will refer to these cultures as human fetal astrocytes.

#### 2.2.2. Brain Endothelial Cells

Human brain endothelial cells (hCMEC/D3 cell line) were cultured as previously described [32]. Briefly, cells were cultured in filtered EBM-2 basal medium supplemented with FBS, hydrocortisone, hFGF-B, VEGF, R3-IGF-1, ascorbic acid, hEGF, and GA-1000 (CC-3202, Lonza Bioscience, Basel, Switzerland) and used for experiments at passages 23–35. The medium was changed every 2–3 days, and the cells were passaged every 4–5 days when they reached 90% confluence. Cells were kept on culture ware coated with 5 μg/cm^2^ rat tail collagen I (Cat no. 122-20, Sigma-Aldrich Merck, Darmstadt, Germany). For all assays (RNA analysis, cell viability, and permeability assays), cells were plated at 50,000 cells per cm^2^ in growth factor-negative medium (EBM-2 with 10 mM HEPES supplemented with FBS, bFGF, hydrocortisone, and ascorbic acid) for 6 days to allow the growth of a confluent monolayer yielding maximum barrier formation.

#### 2.2.3. Cell Culture Treatment

For RNA analysis, human fetal astrocytes originating from four donors were plated in triplicates on PLL-coated 12-well plates and allowed to adhere and grow for 48 h. Brain endothelial cells were plated in quadruplicates on 12-well plates coated with rat tail collagen I and grown for 6 days. Both monocultures were treated with 50 nM PMA (phorbol myristate acetate) and 1000 nM ionomycin (a kind gift from D. Khan and Dr. S. Florquin, Amsterdam UMC, Amsterdam, The Netherlands) in 0.2% DMSO for either 3 h or 6 h. Data from experimental conditions of human fetal astrocytes were normalized to their respective donor control conditions before pooling.

#### 2.2.4. Viability Assay

To determine cell viability of the human fetal astrocytes and endothelial cells after treatment with PMA and ionomycin, as well as after ACT-03 treatment, the MTT (3-(4,5-dimethylthiazol-2-yl)-2,5-diphenyl tetrazolium bromide) assay was performed. After treatment, medium was removed from the cells and replaced with 0.5 mg/mL MTT (Sigma-Aldrich, St Louis, MO, USA) diluted in respective culture medium. After a 1 h incubation at 37 °C in 5% CO_2_, the reaction was halted by aspirating the MTT. Cells were lysed by addition of 100 μL lysis buffer (4 mM HCl, 0.1% Nonidet P-40 in isopropanol). After careful agitation, optical density was measured at 570 nm wavelength using a microplate reader (BMG Labtech, Ortenberg, Germany). Absorbance measurements were normalized to control conditions to visualize relative cell viability. For human fetal astrocytes, cells from four donors were measured in quadruplicates while brain endothelial cells were measured using six replicates.

#### 2.2.5. BBB Model

As an in vitro BBB model, we co-cultured human fetal astrocytes and brain endothelial cells on Transwell^®^ polyester membrane inserts (0.4 µm pores; Cat. No. 3460, Corning, Corning, NY, USA). Briefly, astrocytes (40,000 per cm^2^) were seeded on the bottom of the inserts and allowed to adhere and grow in astrocyte incubation medium. After 48 h, brain endothelial cells were seeded on top with a density of 50,000 cells per cm^2^ as described previously [33], creating a direct contact co-culture. Co-cultures were kept in growth factor-negative EBM-2 medium and were refreshed every other day. On day 6, co-culture was treated with 50 nM PMA and 1000 nM ionomycin in 0.2% DMSO for either 3 h or 6 h.

#### 2.2.6. Real-Time Quantitative Polymerase Chain Reaction (RT-qPCR)

After treatment, cells were washed with PBS five times, after which 700 μL Qiazol Lysis Reagent (Qiagen Benelux, Venlo, The Netherlands) was used to dissociate the cells. Using a phenol/chloroform RNA extraction method, chloroform was thoroughly mixed with Qiazol lysate by vortexing. After centrifugation at 12,000× *g* for 15 min at 4 °C, the aqueous phase was collected. An equal amount of ice-cold isopropanol was added as well as 1 μL of glycogen blue (GlycoBlue, Thermo Fisher Scientific, Waltham, MA, USA). Samples were incubated at −20 °C overnight. Subsequently, samples were centrifugated at 20,000× *g* for 35 min at 4 °C. The RNA-containing pellets were washed twice with ice-cold ethyl alcohol (80%). After complete removal of the washing ethanol, pellets were air-dried for 1 min and dissolved in RNase-free water. A Nanodrop 2000 spectrophotometer at 260/280 nm (Thermo Fisher Scientific, Waltham, MA, USA) was used to determine the concentration and purity of the RNA. 250 ng of RNA was then reverse-transcribed into DNA using oligo-dT primers. PCRs were run on a Roche Lightcycler 480 thermocycler (Roche Applied Science, Basel, Switzerland) using the reference genes elongation factor 1α (EF1-α) in combination with chromosome 1 open reading frame 43 (C1ORF43) for human fetal astrocytes, and in combination with glyceraldehyde 3-phosphate dehydrogenase (GAPDH) for brain endothelial cells. RT-qPCR was run on a 384-well plate; samples were run in duplicates. Every reaction contained 1 μL of the respective cDNA, as well as 2.5 μL SensiFAST SYBR Green NoROX kit (Bioline Reagents Limited, London, UK) and 0.4 μM of forward/reverse primers (listed in Table 1) diluted in water. Negative controls were included for every primer pair and contained water instead of cDNA. RT-qPCR was run starting with initial denaturation at 95 °C for 5 min. In total, 45 cycles were run including denaturation at 95 °C for 15 s, annealing at 65°C for 5 s, and extension at 72 °C for 10 s. Fluorescence measurements, which were acquired via single acquisition mode at 72 °C after each cycle, were quantified using LinRegPCR [34]. For human fetal astrocytes, values were normalized to the 6 h stimulated, untreated condition of the respective donor prior to pooling the four donors.

#### 2.2.7. Permeability Assay

For permeability assays, co-cultures from four independent human fetal astrocyte donors were treated in triplicates. After stimulation, well inserts were transferred to new culture plates containing HBSS in the lower compartment. Inserts were washed three times with prewarmed HBSS, and 20 µM of Lucifer Yellow (Cat. No. L-453, Thermo Fisher Scientific, Waltham, MA, USA) in HBSS was added to the upper compartment. After 1 h of incubation in the dark at 37 °C and 5% CO_2_, duplicate samples were collected from the lower compartment on the receiving plate and added to a black, F-bottom 96-well plate (Cat. No. 655090, Greiner Bio-one, Alphen aan den Rijn, the Netherlands). Lucifer Yellow was detected using a microplate reader at 430 nm/545 nm (excitation/emission) [35].

The apparent permeability coefficients were calculated using the following equation:P_app_ = (Vr × Cf)/(Ci × A × t)

P_app_ = apparent permeability coefficient; Vr = receiver volume (mL); Cf = final receiver concentration (μM); Ci = initial upper compartment concentration (μM); A = membrane area (cm^2^); and t = assay time (seconds).

Duplicate Lucifer Yellow measurements were averaged to represent the apparent permeability per insert. Subsequently, values were normalized to the apparent permeability of the unstimulated condition of their respective human fetal astrocyte donor.

#### 2.2.8. Immunocytochemistry

In order to visualize the cells of the BBB model, inserts were washed three times with PBS and incubated with 4% PFA for 20 min at room temperature. PFA was washed away with PBS, and the membrane containing the fixed cells was removed from the insert. Cells were extracted with 0.1% Triton X-100 in PBS for 5 min at room temperature, and the membrane was blocked using 10% normal goat serum in PBS for 30 min at room temperature. Primary anti-GFAP (polyclonal rabbit 1:500, Z0334, DAKO/Agilent Technologies Netherlands, Amstelveen, the Netherlands) and anti-CD31 (monoclonal mouse, 1:125, clone JC70A, M823, DAKO/Agilent Technologies Netherlands, Amstelveen, the Netherlands) antibodies were added and cells were incubated overnight at 4 °C. After washing three times with PBS, cells were incubated with secondary antibodies (goat anti-rabbit Alexa Fluor 488 and donkey anti-mouse Alexa Fluor 568, Invitrogen/Thermo Fisher Scientific, Waltham, MA, USA) for 2 h at room temperature. Fluorescent microscopy was performed using a confocal microscope (TCS SP8-X, Leica, Son, The Netherlands) and images were processed with ImageJ.

### 2.3. Statistical Analysis

Statistical analysis was performed using GraphPad Prism 8.4. Outliers were identified and excluded using the ROUT method (Q = 1%) after which normal distribution was tested using the D’Agostino-Pearson test of normality. Based on normality, ANOVA or Kruskal–Wallis tests were followed up by unpaired *t*-tests and Mann–Whitney U tests, respectively. A *p*-value < 0.05 was assumed to indicate a significant difference between groups.

## 3. Results

### 3.1. ACT-03 Attenuates Seizure-Induced Astro- and Microgliosis

Immunohistochemical stainings for vimentin, a common marker for activated astrocytes, showed that non-kindled control animals (*n* = 10) lacked vimentin-positive astrocytes in the DG and CA1 (Figure 2A,D). In contrast, vehicle-treated kindled animals displayed a high number of vimentin-positive cells with large and coarse processes, indicative of reactive astrocytes (arrowheads; Figure 2B,E). Kindled animals treated with ACT-03 presented with considerably less vimentin positivity (Figure 2C,F) as compared to vehicle-treated kindled animals. These results were confirmed by quantitative analysis of the vimentin-positive area (Figure 2G–J). Immunohistochemical staining for glial fibrillary acidic protein (GFAP) did not show differences between vehicle- and ACT-03-treated kindled animals (Appendix A).

Comparable results were observed when examining Iba1 stainings, a marker for microglia. Hippocampi of non-kindled control animals contained Iba1-positive cells with a morphology of resting microglia with a relatively small soma and multiple thin processes (arrows, Figure 3A,D). The majority of Iba1-positive microglia in the DG and CA1 of vehicle-treated kindled animals had a typical morphology of activated microglia with few coarse processes (arrowheads, Figure 3B,E), while these were less abundant in ACT-03-treated kindled animals. Quantification of the number of Iba1-positive cells revealed that more microglia were present in the hippocampi of vehicle-treated kindled animals compared to control animals (Figure 3G–J, *p* < 0.05), while less Iba1-positive cells were present in the DG of ACT-03-treated animals compared to vehicle-treated animals (Figure 3G,I; *p* < 0.05).

We examined MMP9 protein expression using immunohistochemistry in neurons and glia of the hippocampus and observed higher MMP9 expression in glial cells of the dentate gyrus in kindled rats compared to non-kindled rats. MMP9 expression did not differ between vehicle-treated kindled rats and ACT-03-treated kindled rats (Appendix A).

### 3.2. ACT-03 Reduces the Expression of Various Pro-Inflammatory Factors In Vitro

We further investigated the potential effects of ACT-03 on inflammation in vitro, focusing on its effects in astrocyte cultures. For this purpose, we stimulated human fetal astrocytes with PMA and ionomycin (hereafter referred to as PMA + iono) for either 3 h or 6 h in the absence or presence of ACT-03. PMA + iono stimulation induced overexpression of inflammatory factors interleukin (IL) 1β (Figure 4A; *p* < 0.0001), IL-6 (Figure 4B; *p* < 0.0001), tumor necrosis factor α (TNFα; Figure 4C; *p* < 0.0001), cyclo-oxygenase 2 (COX2; Figure 4D; *p* < 0.0001), transforming growth factor β after 3 h stimulation (TGFβ; Figure 4E; 3 h, *p* < 0.01) and its receptor 2 (TGFβ-R2) as measured 6 h after stimulation (Figure 4F; 6 h, *p* < 0.001). In the presence of ACT-03, PMA + iono stimulation resulted in a lower mRNA expression of IL-1β (3 h, *p* < 0.05; 6 h, *p* < 0.0001), IL-6 (6 h, *p* < 0.001) and TGFβ-R2 (6 h, *p* < 0.05) as compared to the condition where only PMA + iono was present.

We also investigated the expression of the aforementioned inflammatory factors in the human brain endothelial cell line hCMEC/D3. PMA + iono stimulation in absence of ACT-03, resulted in higher mRNA expression of TNFα (Figure 5C; *p* < 0.05), TGFβ (Figure 5E; *p* < 0.05), and TGFβ receptor 1 (TGFβ-R1; Figure 5F; *p* < 0.05), and a trend towards higher mRNA expression of IL-1β (Figure 5A, *p* = 0.0571), IL-6 (Figure 5B; *p* = 0.0571) and COX2 (Figure 5D; *p* = 0.0571). In presence of ACT-03, overexpression of TGFβ after 6 h stimulation and TGFβ-R1 after 3 h stimulation was no longer observed. Furthermore, PMA + iono stimulation resulted in higher mRNA expression of intercellular adhesion molecule 1 (ICAM1, Figure 5G, *p* < 0.05), as well as a lower mRNA expression of platelet endothelial cell adhesion molecule (PECAM1/CD31; Figure 5H; *p* < 0.05) compared to vehicle-treated cells. In the presence of ACT-03, PMA + iono stimulation led to higher expression of ICAM1, while PECAM1 expression did not change. Interestingly, ACT-03 treatment concurrent with PMA + iono stimulation compared to stimulation alone resulted in lower expression of IL-6 (6 h, *p* < 0.05), COX2 (*p* < 0.05) and ICAM1 (*p* < 0.05), and in a trend towards lower expression of TGFβ (6 h, *p* = 0.0571) and TGFβ-R1 (6 h, *p* = 0.0571). Collectively, these data show that ACT-03 is able to downregulate several cyto- and chemokines in a neuroinflammatory environment in vitro.

### 3.3. Effects of ACT-03 on the Expression of MMPs and Transcriptional Regulators

In order to investigate whether ACT-03 could also affect the proteins it specifically targets, we examined the expression of MMP2 and nine transcripts, along with that of their family member MMP3, which also plays an important role in neuroinflammatory processes [30,36]. In astrocytes, independent of ACT-03, higher expression of MMP3 and MMP9 was observed after 3 h and 6 h of PMA + iono stimulation compared to their respective non-stimulated controls (Figure 6B,C; *p* < 0.0001). MMP2 mRNA expression, however, showed a modest decrease after 6 h of stimulation in the absence of ACT-03. Compared to PMA + iono alone, treatment with ACT-03 resulted in lower expression of MMP3 and MMP9 mRNA after 6 h (*p* < 0.0001, *p* < 0.05, respectively). Stimulation of endothelial cells yielded similar results: lower expression of MMP2 (Figure 7A; 6 h, *p* < 0.05) and higher MMP3 (Figure 7B; *p* < 0.05) and MMP9 expression (Figure 7C; *p* < 0.05). Similarly, in presence of ACT-03, PMA + iono stimulation led to lower MMP2 expression (6 h, *p* < 0.05) and higher MMP9 expression (*p* < 0.05). Higher MMP3 expression, however, was no longer observed. Additionally, when comparing the treatment groups, lower MMP9 expression was observed (6 h, *p* < 0.05) as well as a trend towards lower MMP3 expression (6 h, *p* = 0.0571).

Both MMPs, as well as several aforementioned cyto- and chemokines, are transcribed under the control of transcription factor activator protein (AP) 1, among others [37,38]. We therefore examined the mRNA expression of the AP-1 subunits cFOS and cJUN, and observed that in both cell types, cFOS expression was higher after stimulation in the presence and absence of ACT-03 compared to the non-stimulated condition (Figure 6D, astrocytes, *p* < 0.0001; 7D, brain endothelial cells, *p* < 0.05). Similarly, cJUN expression was higher after 3 h of PMA + iono stimulation (astrocytes, *p* < 0.01, brain endothelial cells, *p* < 0.05). Between treatment comparisons showed that the cFOS expression was lower in both astrocytes (3 h, *p* < 0.05) and brain endothelial cells (*p* < 0.05) that were incubated with ACT-03. These results indicated that besides the suppression of MMP activation, ACT-03 seems to have an additional transcriptional effect on MMPs, possibly via the downregulation of transcription factor AP-1.

### 3.4. Modulatory Effect of ACT-03 on In Vitro Barrier Integrity

Considering the effectiveness of ACT-03 in MMP inhibition [27,28] and its ability to downregulate several pro-inflammatory markers, we wondered whether ACT-03 treatment could also have beneficial effects on endothelial integrity. In the rapid kindling rat model, albumin immunohistochemistry did not show extravasation of the blood-borne protein albumin into the parenchyma 9 days after the last kindling session (Appendix A). Therefore, we sought to use a BBB in vitro model consisting of co-cultured brain endothelial cells and astrocytes (Figure 8A,B). The integrity of the in vitro barrier was tested by determining the apparent permeability (P_app_) to the ±450 kDa protein Lucifer Yellow provided to the apical chamber of a Transwell^®^ insert. The average P_app_ for Lucifer Yellow in naïve conditions was 8.06 × 10^−6^ cm/s (SEM = 6.28 × 10^−7^), which is considered a representative barrier integrity under normal physiological conditions [39,40,41]. PMA + iono stimulation of the co-culture resulted in higher barrier permeability after 6 h (Figure 8D, *p* < 0.05). This increase was not observed in the presence of ACT-03. Instead, addition of ACT-03 resulted in a lower of barrier permeability (*p* < 0.01). To examine whether the change in barrier permeability was caused purely by changes in para- or pericellular transport or partially due to loss of cells, a cell viability assay was performed for both cell types. Astrocytes (Figure 8E) and brain endothelial cells (Figure 8F) responded to PMA + iono stimulation similarly in both the absence as well as in the presence of ACT-03. After PMA + iono stimulation, cell viability was lower in astrocytes after 3 h and 6 h of stimulation, with 18.5% and 13.6%, respectively (Figure 8E, *p* < 0.01). In the presence of ACT-03, a similar pattern was observed, as cell viability decreased to 19.3% after 3 h stimulation (*p* < 0.01) which recovered at 6 h after stimulation (7.3%, *p* > 0.05). Interestingly, in the absence of ACT-03, brain endothelial cell viability was higher after 6 h of stimulation (Figure 4F; 15.6%; *p* < 0.001), comparable to an 11.3% (*p* > 0.05) in the presence of ACT-03. Altogether, these data indicate that ACT-03 is able to have a protective effect on barrier integrity.

## 4. Discussion

We have shown that astro- and microgliosis was less in the brain of kindled rats after treatment with ACT-03, compared to vehicle-treated animals. Moreover, induced overexpression of several cytokines, chemokines, and matrix metalloproteinases and their transcription factors was reduced by ACT-03 treatment in both human fetal astrocytes as well as human brain endothelial cell cultures. Furthermore, ACT-03 treatment had a protective effect on barrier integrity.

### 4.1. Astroglial and Inflammatory Responses Dampened by ACT-03

Persistent activation of astrocytes and microglia may lead to a pro-inflammatory environment, which has been implicated in the development of many neurological disorders including several types of epilepsies [42,43]. Activation of astrocyte and microglia and increased expression of their signature markers has been shown after brain insults such as a status epilepticus, induced seizures or brain trauma [43,44,45,46,47]. Staining for the commonly used marker for astrogliosis, GFAP, did not show a difference between vehicle-treated and ACT-03-treated kindled rats. However, as GFAP is a highly abundant protein and the hippocampus has a high constitutive presence of GFAP-positive astrocytes, GFAP quantification might not best represent reactive astrogliosis [48]. Hence, we evaluated astrogliosis using another commonly used marker for reactive astrocytes, vimentin [49,50]. Evaluating the vimentin-positive astrocytes and Iba1-positive microglia, we observed astrogliosis and microgliosis in the hippocampus of kindled rats. Morphologically, these glial cells had enlarged somata and coarse processes, indicative of a reactive state. Interestingly, ACT-03 treatment reduced these morphological changes. Decreased expression of reactive glial markers has been observed in multiple disease models utilizing MMP inhibitors [51,52,53,54,55,56,57]. MMP inhibitor treatment often goes along with a decrease in injury and/or symptoms, which was also observed in our previous study where ACT-03 (IPR-197) increased resistance to kindling, reduced seizure severity according to Racine’s scale [58], seizure frequency, and seizure duration after kindling [28]. This dampened kindling effect in ACT-03-treated animals compared to controls may also explain a reduced glial response, although the interdependence of gliosis and temporal lobe epilepsy is still debated [59]. We did not observe a decrease in MMP9 protein expression in ACT-03-treated rats. However, as ACT-03′s mode of action is through inhibition of the protein [27,28], not reduced translation, this is not an indication of ACT-03 efficacy. To more specifically study ACT-03′s effects on inflammatory responses, we induced an inflammatory state in human fetal astrocyte cultures and investigated the effects of ACT-03 on the mRNA expression of various pro-inflammatory factors.

To model the environment of a pro-inflammatory state, we used PMA and ionomycin stimulation of astrocytes and endothelial cells, which has previously been shown to upregulate inflammatory mediators relevant for mimicking the changes seen during epileptogenesis [60,61,62,63]. PMA + iono treatment caused the upregulation of several chemo- and cytokines (IL-1β, IL-6, COX2, TGFβ, and its receptor), as well as MMP3 and MMP9. Simultaneous treatment with ACT-03 resulted in lower mRNA expression of several of these markers, including interleukins. Interleukin 1 is considered a master cytokine of local and systemic inflammation and has been implicated in several malfunctioning brain processes and neurological disorders [64,65,66]. The binding of IL-1β to its receptor elicits activation of the transcription factor NF-κB and the mitogen-activated protein kinase signaling pathway. These two pathways cause downstream transcription and activation of several pro-inflammatory factors, enhancing the neuroinflammatory milieu [67,68,69,70]. Likewise, IL-6 plays an important role in glial activation during brain injury [71,72] and its increased expression is often observed in several neurological disorders [71,73]. IL-6 seems to have a neuroprotective effect in both physiological and early responses [74,75] and a neurotoxic effect in progressed disease states [76]. Furthermore, a meta-analysis identified IL-6 in blood plasma associated with poor clinical outcome after stroke [77], and elevated levels of IL-6, in particular, are considered characteristic of cytokine storms [78]. Taken together, the reduction by ACT-03 of increased interleukin expression upon stimulation indicates a dampening of the inflammatory response of the cells. Similar to IL-6, TGFβ is a versatile cytokine involved in several processes including cell proliferation and migration, angiogenesis [79,80], and gliogenesis [81,82,83]. While TGFβ can have anti-inflammatory properties [84,85], overexpression of TGFβ in astrocytic cells in vivo resulted in hippocampal gliosis, possibly via the activation of the GFAP promotor [86,87]. Upregulation of TGFβ can be found in patients with epilepsy, Alzheimer’s disease, and others, and its inhibition has been shown to reduce seizures in models of acquired epilepsy [88]. In this study, we observed an increase of both TGFβ and its receptor TGFβ-R2 in astrocytes, while ACT-03 treatment reduces TGFβ-R2. Thereby, ACT-03 might diminish downstream signaling of TGFβ and the associated inflammatory changes, including gliosis [89].

### 4.2. ACT-03 Treatment Partially Ameliorates Endothelial Activation

Endothelial cells are known to play a role in local inflammatory processes and respond to, as well as synthesize, various inflammatory factors [90,91]. After treatment with ACT-03, we observed a decrease of stimulation-induced IL-6 expression in human brain endothelial cell cultures. Interestingly, IL-6 in brain endothelial cells can directly and indirectly cause injury that could result in vascular leakage [92]. Additionally, it can also cause release of other cytokines such as IL-8 and MCP-1, aggravating brain inflammation [93]. We also observed higher expression of COX2 in endothelial cells after stimulation. Since COX2 is known to be induced in endothelial cells by inflammatory stimuli and shear stress [94,95], and is involved in the development of vascular inflammation [96,97], the subsequent decrease of COX2 and IL-6 after ACT-03 treatment might be an indication of the dampened activation state of these cells.

Additionally, ACT-03 was able to reduce the stimulation-induced increase in TGFβ and its receptor TGFβ-R1 in endothelial cells. The directionality of the effect of endothelial TGFβ is highly context-dependent and can be both in favor of as well as opposed to angiogenesis and BBB integrity [83]. Moreover, the downstream effects of TGFβ-R1 binding depend on the type of receptor involved, Alk1 or Alk5 [98,99]. While the former leads to promotion of endothelial cell migration and proliferation, the latter leads to a quiescent state of the cells and stabilization of the barrier. Despite that, a body of evidence points towards the role of TGFβ in BBB dysfunction [100,101,102], and future studies should aim to unravel whether either of TGFβ-R1′s downstream pathways might be MMP9 dependent. In line with the protective effects of MMP inhibition, we also observed that higher ICAM1 and lower PECAM1 expression due to stimulation were restored after ACT-03 treatment. As dysregulation of these adhesion molecules is suggestive of an enhanced cellular activation and reduced barrier integrity [103,104,105], these results might point towards barrier-protecting effects of ACT-03.

### 4.3. Reduced MMP Transcription Potentially through Indirect Negative Feedback

In order to examine whether the reduction of activation in astrocytes and endothelial cells after ACT-03 treatment extended to the expression of MMPs themselves, we measured MMP2, MMP3, and MMP9 mRNA. Interestingly, in both cell types, MMP3 and MMP9 overexpression was reduced after ACT-03 treatment. ACT-03′s mode of action is through binding to the highly conserved active site of MMP2 and MMP9 [27]. Thereby, it decreases the enzyme’s activity, impeding the enzyme of cleaving its intended targets. We have previously shown that ACT-03 is highly specific for MMP2 and MMP9 and does not react with other peptidases [27]. It is therefore unlikely that the effects of ACT-03 on mRNA expression of inflammatory mediators and adhesion molecules are direct effects. MMPs as well as several other inflammatory mediators are under the transcriptional control of transcription factor AP-1 [106,107,108,109,110]. We observed that ACT-03 treatment reduced mRNA expression of the AP-1 subunit cFOS. Moreover, it is known that MMPs are major contributors to neuroinflammatory processes. Their degradome includes several pro-forms of cytokines, such as pro-IL1β and pro-TNFα [111,112,113]. MMPs are responsible for the removal of the pro-peptides, thereby activating these proteins. Additionally, as MMPs have the ability to cleave and breakdown several extracellular matrix components, uncontrolled peptidase activity might lead to destabilization of extra- and intracellular homeostasis [114,115] and subsequent upregulation of inflammatory mediators. ACT-03 dampens this cleavage activity and could thereby soothe the cell’s response of increased transcription of abovementioned factors, including MMPs themselves.

### 4.4. Functional Beneficial Effects of ACT-03 on Barrier Integrity

Finally, to investigate the potential beneficial effects of ACT-03 on barrier integrity, we examined the presence of the blood protein albumin in the brains of kindled rats. Albumin immunoreactivity was only observed within the vasculature, whereas parenchymal albumin was not present. We cannot exclude the possibility that BBB dysfunction occurred during rapid kindling; however, considering that animals were sacrificed more than a week after the last rapid kindling session, and only had one single, less intense re-kindling session afterwards, the BBB might have been restored over that time. Since BBB dysfunction is observed in other rodent models of epilepsy, as well as in patients [116,117,118,119,120,121], we were interested in studying the effect of ACT-03 on barrier function, and employed a functional assay by combining astrocyte and endothelial cultures in an in vitro BBB model. ACT-03 was able to rescue the stimulation-induced loss of barrier integrity and restored barrier permeability back to control levels. Stimulation by PMA and ionomycin seemed to have a slight proliferative effect on endothelial cells, which might have been involved in destabilization of the barrier. However, as a comparable change was observed after ACT-03 treatment, similar effects of cell proliferation would be present. Nevertheless, ACT-03 displayed a rescuing effect on barrier permeability, indicating that there is restored control of paracellular transport.

It has been shown that MMPs are directly involved in the dysregulation of BBB integrity. Among their many substrates, MMPs target multiple tight junction proteins for degradation, such as ZO-1, occluding, and claudins [122]. These molecules mediate the controlled transport through the endothelial layer of the BBB and are responsible for the vascular integrity of capillaries [123]. Moreover, MMPs are known to target ECM components of the basement membrane found between the endothelial cells, pericytes, and astrocytic endfeet. As the basement membrane serves simultaneously as a physical scaffold and a barrier, loss of ECM molecules by MMP activity, including collagen, laminins, fibronectin, and heparin sulphates, can lead to instability and increased BBB permeability [9,15]. Indeed, several in vitro and in vivo studies have shown that MMPs, and most often MMP9, can cause disruption of the BBB [13,14,124]. Likewise, inhibition of MMPs has been shown to limit BBB disruption by preventing the loss of tight junction proteins [125,126]. In fact, MMP inhibition has been proposed as treatment of several brain disorders [9,22]. However, previously used MMP inhibitors do not show specificity for MMP2 and MMP9, and treatment with broad spectrum MMP inhibitors has been proven to induce side effects in phase I and II oncological clinical trials [22]. Our study shows that specific targeting of MMP2 and MMP9 by ACT-03, however, has beneficial effects in animal models of focal temporal lobe epilepsy without side effects [28], thus providing a novel and interesting pharmacologic alternative for the treatment of brain disorders with neuroinflammatory components.

## 5. Conclusions

Our results show that ACT-03 reduces astro- and microgliosis in the rapid kindling model, and attenuates inflammation and loss of endothelial integrity in vitro. These data mechanistically support the antiseizure and antiepileptogenic effects previously reported in two rodent models of temporal lobe epilepsy. Therefore, ACT-03 is of interest for further clinical evaluation as a novel therapeutic treatment for epilepsies with brain inflammation and/or blood–brain barrier dysfunction.

## 6. Patents

RP is an employee of Accure Therapeutics SL, which has the patent WO 2017/085034 A1, “Gelatinase inhibitors and use thereof”.

## Figures and Tables

**Figure 1 biomedicines-10-02117-f001:**
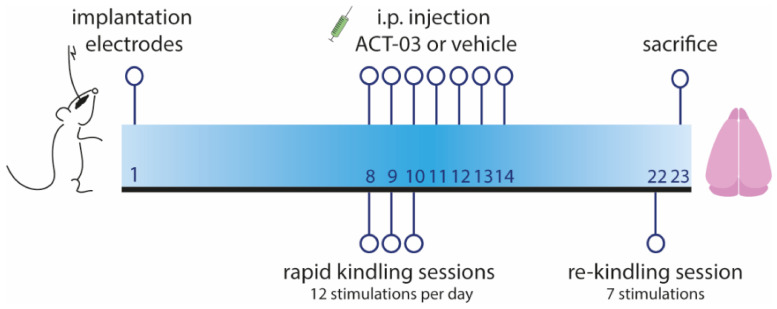
Timeline of rapid kindling rat experiment. One week after implantation of stimulation and EEG electrodes in the angular bundle and hippocampus respectively, animals were given 12 kindling stimulations per day for three consecutive days. Starting on the first day of stimulation, animals were treated intraperitoneally (i.p.) with vehicle or ACT-03 (one hour before the first stimulation), continuing for one week in total. After one week, in absence of the drug, a re-kindling session (with seven stimulations) was performed. One day later, animals were sacrificed and the brain was collected for further processing, as described in Appendix A and by Broekaart et al. 2021 [28].

**Figure 2 biomedicines-10-02117-f002:**
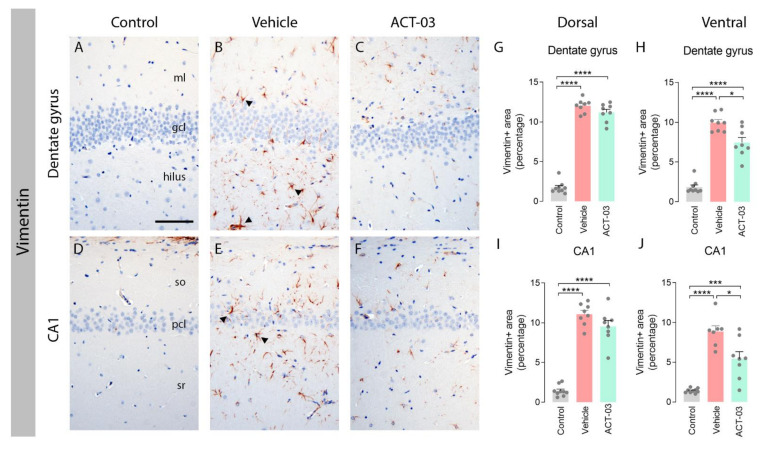
Increased astrogliosis after kindling is ameliorated by ACT-03 treatment. (**A**–**F**) In non-kindled control animals, vimentin immunoreactivity (IR) was not observed in the parenchyma (**A**,**D**), while vimentin-positive astrocytes with coarse processes and a reactive morphology (arrowheads) were observed in the hippocampus of kindled, vehicle-treated animals (**B**,**E**). In kindled animals treated with ACT-03, vimentin-positive cells were sparsely present and displayed less of a reactive morphology (**C**,**F**). (**G**–**J**) Quantification shows a larger vimentin-positive area in the dorsal and ventral dentate gyrus (DG) and CA1 area of kindled animals compared to non-kindled animals (*p* < 0.001). The vimentin-positive area in kindled, ACT-03-treated animals was smaller compared to kindled, vehicle-treated animals in the DG and CA1 of the dorsal hippocampus (*p* < 0.05). Scale bar in A for A–F; 100 μm; hematoxylin counterstained; ml, molecular later; gcl, granular cell layer; so, stratum oriens; pcl, pyramidal cell layer; sr, stratum radiatum. Bar graphs display mean + SEM, with individual values as dots; *, *p* < 0.05; ***, *p* < 0.001; ****, *p* < 0.0001.

**Figure 3 biomedicines-10-02117-f003:**
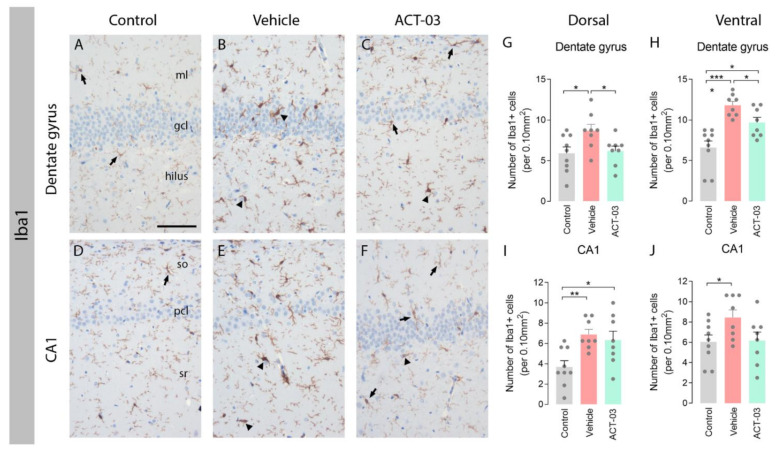
Increased microgliosis after kindling is ameliorated by ACT-03 treatment. (**A**–**F**) Iba1-positive microglia with a resting morphology (arrows) were observed in the hippocampus of non-kindled animals (**A**,**D**), where Iba1-positive microglia in kindled, vehicle-treated animals have thick and coarse processes (arrowheads). In kindled, ACT-03-treated animals, Iba1-positive microglia with both resting and active morphologies were observed (**C**,**F**). (**G**–**J**) Quantification shows a higher number of Iba1-positive microglia in kindled versus non-kindled animals in multiple regions (*p* < 0.05). Iba1-positive cells was lower in the DG of the dorsal and ventral hippocampus of ACT-03-treated animals versus vehicle-treated animals. Scale bar in A for A–F; 100μm; hematoxylin counterstained; ml, molecular later; gcl, granular cell layer; so, stratum oriens; pcl, pyramidal cell layer; sr, stratum radiatum. Bar graphs display mean + SEM, with individual values as dots; *, *p* < 0.05; **, *p* < 0.01; ***, *p* < 0.001.

**Figure 4 biomedicines-10-02117-f004:**
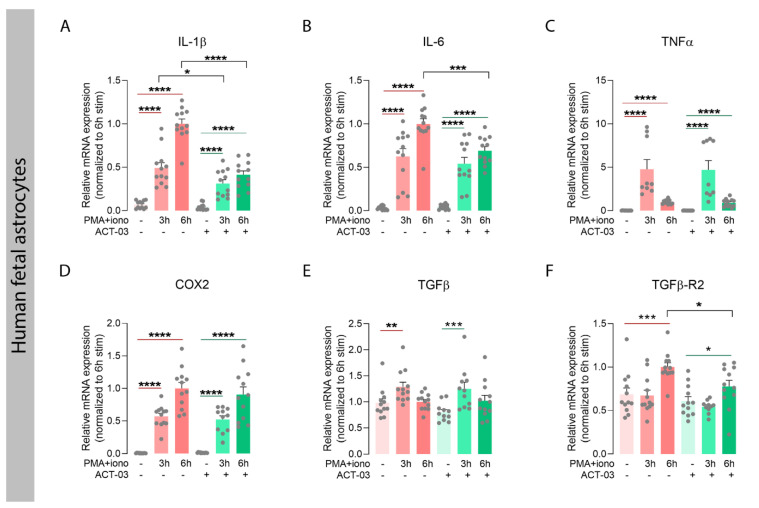
Attenuation of inflammatory factors by ACT-03 in human fetal astrocytes. (**A**–**F**) Quantitative PCR of inflammatory mediators shows that stimulation of human fetal astrocytes (*n* = 4 donors in experimental triplicates) increases relative mRNA expression of interleukin 1 beta (IL-1β), IL-6, tumor necrosis factor alpha (TNFα), cyclo-oxygenase 2 (COX2), transforming growth factor beta (TGFβ) and TGFβ receptor 2 (TGFβ-R2) in response to 3 h and/or 6 h stimulation with 50 nM phorbol myristate acetate (PMA) and 1000 nM ionomycin (iono) (*p* < 0.01). Simultaneous treatment with 150µM ACT-03 resulted in a decrease mRNA expression IL-1β, IL-6, and TGFβ-R2 compared to PMA + iono stimulation solely (*p* < 0.05). Bar graphs display mean + SEM, with individual values as dots; *, *p* < 0.05; **, *p* < 0.01; ***, *p* < 0.001; ****, *p* < 0.0001. Within treatment comparisons indicated with red/green lines. Between treatment comparisons indicated with hooked black lines.

**Figure 5 biomedicines-10-02117-f005:**
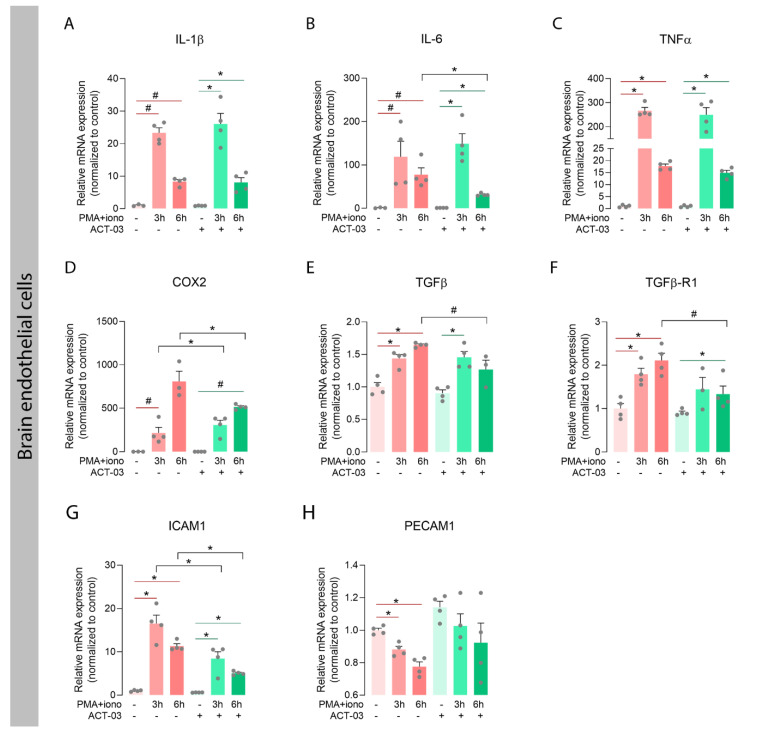
Attenuation of inflammatory factors by ACT-03 in brain endothelial cell cultures. (**A**–**H**) PMA + iono stimulation of the human brain endothelial cell line hCMEC/D3 (*n* = 4) induced higher expression of TNFα, TGFβ, TGFβ-R1, and ICAM1 (*p* < 0.05), and trends towards higher expression of IL-1β, IL-6, and COX2 (*p* = 0.0571). ACT-03 treatment reduced the PMA + iono-induced expression of IL-6, COX2, and ICAM1 (*p* < 0.05) and resulted in a trend towards less expression of TGFβ and TGFβ-R1 (*p* = 0.0571). Bar graphs display mean + SEM, with individual values as dots; ^#^, *p* = 0.0571; *, *p* < 0.05. Within treatment comparisons indicated with red/green lines. Between treatment comparisons indicated with hooked black lines.

**Figure 6 biomedicines-10-02117-f006:**
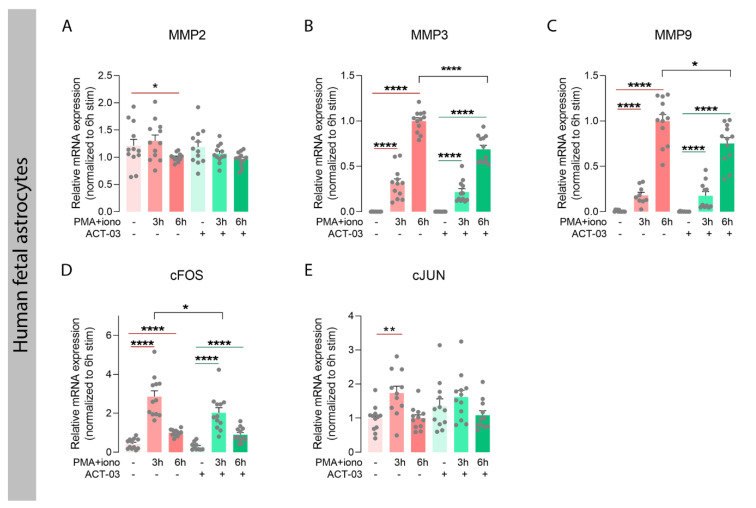
Attenuation of metalloproteinases and inflammatory transcription factors by ACT-03 in astrocytes. (**A**–**E**) Relative mRNA expression of human fetal astrocytes (*n* = 4 donors in experimental triplicates) measured by quantitative PCR showed less matrix metalloproteinase 2 (MMP2) and higher expression of MMP3, MMP9, and subunits of the transcription factor activator protein 1, cFOS, and cJUN, after 3 and/or 6 h stimulation with 50 nM phorbol myristate acetate (PMA) and 1000 nM ionomycin (iono) (*p* < 0.01). Simultaneous treatment with 150 µM ACT-03 resulted in lower mRNA expression of MMP3, MMP9, and the cFOS gene compared to stimulation only (*p* < 0.05). Bar graphs display mean + SEM, with individual values as dots; *, *p* < 0.05; **, *p* < 0.01; ****, *p* < 0.0001. Within treatment comparisons indicated with red/green lines. Between treatment comparisons indicated with hooked black lines.

**Figure 7 biomedicines-10-02117-f007:**
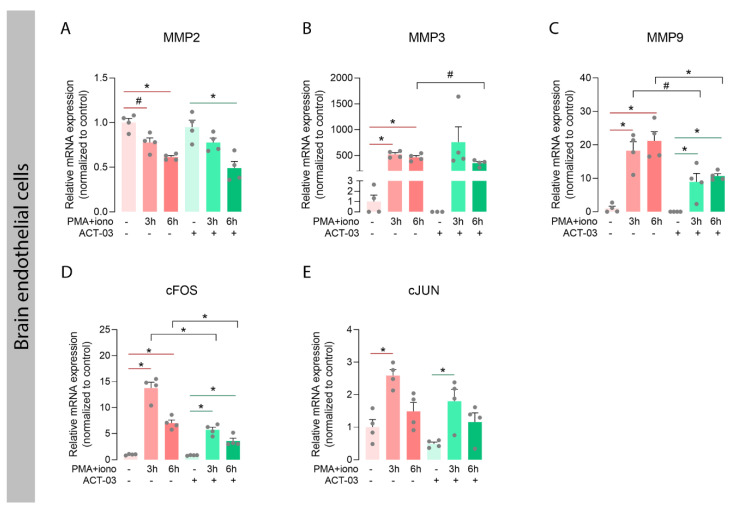
Attenuation of metalloproteinases and inflammatory transcription factors by ACT-03 in brain endothelial cells. (**A**–**E**) In the human brain endothelial cell line hCMEC/D3 (*n* = 4), PMA + iono stimulation resulted in less MMP2 expression and higher expression of MMP3, MMP9, cFOS, and cJUN (*p* < 0.05). Upon ACT-03 treatment, less cFOS expression was observed compared to stimulation only (*p* < 0.05), as well as a trend towards less MMP3 and MMP9 expression (*p* = 0.0571, *p* < 0.05). Bar graphs display mean + SEM, with individual values as dots; #, *p* = 0.0571; *, *p* < 0.05. Within treatment comparisons indicated with red/green lines. Between treatment comparisons indicated with hooked black lines.

**Figure 8 biomedicines-10-02117-f008:**
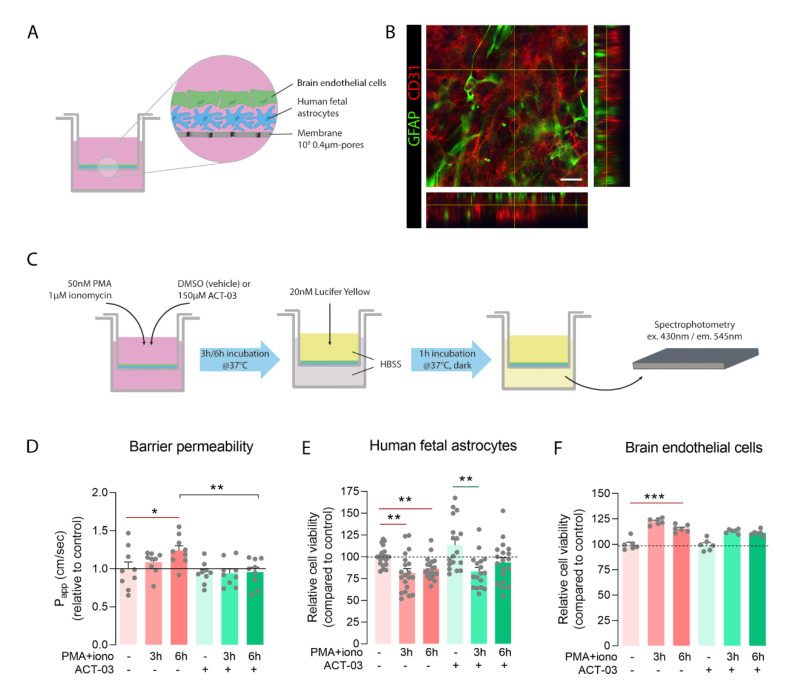
ACT-03 treatment rescues loss of barrier integrity in vitro. (**A**) Schematic representation of the in vitro blood–brain barrier (BBB) model containing human fetal astrocytes and human brain endothelial cells (hCMEC/D3) plated on a Transwell^®^ insert with 0.4µm-sized pores. (**B**) Immunofluorescent staining of the membrane shows GFAP-positive astrocytes (red) under a monolayer of CD31-positive endothelial cells (green) of. (**C**) Scheme of experimental layout showing the stimulation of the in vitro BBB model with phorbol myristate acetate (PMA) and ionomycin in the presence of either vehicle or ACT-03 for 3 h or 6 h. Subsequently, medium was exchanged for Hank’s balanced salt solution (HBSS) and 20 nM of Lucifer Yellow (LY) was added to the upper compartment of the insert. After 1 h of incubation, LY absorbance of the lower compartment of the Transwell^®^ was measured in triplicates using spectrophotometry. (**D**) Higher barrier permeability indicated by apparent permeability (P_app_) was observed after 6 h of stimulation (*p* < 0.05). Addition of ACT-03 reduced barrier permeability compared to stimulation alone (*p* < 0.01), returning values to control level. (**E**) MTT assays of human fetal astrocytes and (**F**) endothelial cells show cell viability after PMA + iono stimulation. Monocultures of human fetal astrocytes were performed on *n* = 4 donors in 4 experimental replicates, endothelial cell cultures were performed with 6 experimental replicates. The co-culture in vitro BBB model was performed using *n* = 3 astrocytes donors with experimental triplicates. Scale bar: 12.5 μm. Bar graphs display mean + SEM, with individual values as dots; *, *p* < 0.05; **, *p* < 0.01; ***, *p* < 0.001. Within treatment comparisons indicated with red/green lines. Between treatment comparisons indicated with hooked black lines.

**Table 1 biomedicines-10-02117-t001:** Primers used for quantitative real-time PCR.

*Gene*	*Name*	*Forward*	*Reverse*
** *EF1α* **	Elongation factor 1 alpha	ATCCACCTTTGGGTCGCTTT	CCGCAACTGTCTGTCTCATATCAC
** *C1orf43* **	Chromosome 1 open reading frame 43	GATTTCCCTGGGTTTCCAGT	ATTCGACTCTCCAGGGTTCA
** *GAPDH* **	Glyceraldehyde 3-phosphate dehydrogenase	AGGCAACTAGGATGGTGTGG	TTGATTTTGGAGGGATCTCG
** *IL-1β* **	Interleukin 1 beta	GCATCCAGCTACGAATCTCC	GAACCAGCATCTTCCTCAGC
** *IL-6* **	Interleukin 6 beta	CTCAGCCCTGAGAAAGGAGA	TTTCAGCCATCTTTGGAAGG
** *TNFα* **	Tumor necrosis factor alpha	CCCCAGGGACCTCTCTCTAA	CAGCTTGAGGGTTTGCTACA
** *COX2* **	Cyclo-oxygenase 2	GAATGGGGTGATGAGCAGTT	GCCACTCAAGTGTTGCACAT
** *TGFβ* **	Transforming growth factor beta	GTGGAAACCCACAACGAAAT	CGGAGCTCTGATGTGTTGAA
** *TGFβ-R1* **	Transforming growth factor beta receptor 1	AAGAACGTTCGTGGTTCCGT	CACCAACCAGAGCTGAGTCC
** *TGFβ-R2* **	Transforming growth factor beta receptor 2	CCACCGCACGTTCAGAAGTC	GTCCTATTACAGCTGGGGCA
** *ICAM1* **	Intracellular adhesion molecule 1	AGCTTCGTGTCCTGTATGGC	TTTTCTGGCCACGTCCAGTT
** *PECAM1* **	Platelet and endothelial cell adhesion molecule 1 (CD31)	GGAAGTTCAAGTGTCCTCAGC	GGAGCCTTCCGTTCTAGAGT
** *MMP2* **	Matrix metalloproteinase 2	ATAACCTGGATGCCGTCGT	AGGCACCCTTGAAGAAGTAGC
** *MMP3* **	Matrix metalloproteinase 2	CTCCAACCGTGAGGAAAATC	CATGGAATTTCTCTTCTCATCAAA
** *MMP9* **	Matrix metalloproteinase 3	GAACCAATCTCACCGACAGG	GCCACCCGAGTGTAACCATA
** *cFOS* **	Activator protein-1 subunit cFOS	GGAGAATCCGAAGGGAAAGGA	GCAAAGCAGACTTCTCATCTTCT
** *cJUN* **	Activator protein-1 subunit cJUN	CCAACTCATGCTAACGCAGC	TCTCTCCGTCGCAACTTGTC

## Data Availability

Not applicable.

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
