# Peer review of "The Gelatinase Inhibitor ACT-03 Reduces Gliosis in the Rapid Kindling Rat Model of Epilepsy, and Attenuates Inflammation and Loss of Barrier Integrity In Vitro"

_biomedicines, 2022, doi:10.3390/biomedicines10092117_

Round 1

Reviewer 1 Report

The submission from Diede W.M. Broekaart et al. reports that the gelatins inhibitor ACT-03 reduces gloss in the rapid kindling model of epilepsy attenuating the inflammation and the loss of barrier integrity. The study is interesting and well structured. However, there are some corrections.

Minor comments:

1.     Please refer to doi: 10.1177/1535759720944924; 10.1016/j.nbd.2019.01.007

2. Line 158: the authors write figure 12. Is this an error?

3.  The authors should include in the figure 1 the scale bar.

3.     The authors should better highlight the purpose of the study and the novelty

4.     The authors should better check the manuscript for any typographical errors.

Author Response

The submission from Diede W.M. Broekaart et al. reports that the gelatins inhibitor ACT-03 reduces gloss in the rapid kindling model of epilepsy attenuating the inflammation and the loss of barrier integrity. The study is interesting and well structured. However, there are some corrections.

Minor comments:

Point 1: Please refer to doi: 10.1177/1535759720944924; 10.1016/j.nbd.2019.01.007

Response 1: These references are included now.

Point 2: Line 158: the authors write figure 12. Is this an error?

Response 2: Thank you for pointing this out. This has now been corrected.

Point 3: The authors should include in the figure 1 the scale bar.

Response 3: The scale bar in Figure 1A is representative for all panels in that figure as mentioned in the figure legend. For clarity, we have also added the scale bar to panel K.

Point 4: The authors should better highlight the purpose of the study and the novelty

Response 4: The novelty of this manuscript lies in the understanding of the mechanism of actions of a novel MMP inhibitor in order to interpret its potential in other types of neurological disorders where overlapping pathological processes are present. We have more clearly stated this in the introduction.

Point 5: The authors should better check the manuscript for any typographical errors.

Response 5: We have re-read the manuscript and corrected these errors.

Reviewer 2 Report

we read with interest the article by  Broekaart et al entitled "The gelatinase inhibitor ACT-03 reduces gliosis in the rapid kindling model of epilepsy, and attenuates inflammation and loss of barrier integrity in vitro" where the authors aimed at assessing the novel gelatinase inhibitor ACT-18 03 as a disease-modifying drug in models of epilepsy via evaluation of neuroinflammation and by trying to test its neuroprotective effects by preserving the blood-brain barrier (BBB) integrity and reducing inflammation in vitro.

There are major comments on this work, the authors have invested in the epilepsy model and only assessed inflammation in the brain with iba1 and vimentin (which should be replaced with GFAP as a better marker for astrogliosis) and then they reverted to cell culture models using ionomycin and PMA!!! Which lowers the value of the discussed drug ACT-03 drug. Also, NeuN should be evaluated to show the lesser neural injury is occurring

First, the BBB should be evaluated in the rat model used and the ionomycin and PMA treatment for the in vitro BBB treated with ACT-30 drug represent o evidence for protection for BBB integrity.

The animal model should be  the primary aim for the drug rather than the artificial cell culture model that relied mainly on rt-PCR which also doesn’t reflect the neuroprotection of the proposed drug

The MMP-2 and 9 should be assessed for their expression and activity on the protein level rather than on the gene level which again, has lesser value being performed on the cell culture rather than on animals

The same applies to the pro-inflammatory cytokines being assessed via RT-PCR and reflected minor changes.

Based on these comments, I request that the cell culture data be repeated on the animal model to determine any value for the ACT-03 drug. I don’t think there is much scientific merit in in-vitro experiments.

Minor comments:

1.     The authors should discuss the value of using ionomycin and PMA

2.      What is meant by n=10 in the method section “Kindled rats were treated daily intraperitoneally (i.p.) with either vehicle (5.0 % 88 Tween® 80 in 0.9 % saline; n=10) or 6 mg/kg ACT-03 (n=10, Accure Therapeutics”

Author Response

we read with interest the article by  Broekaart et al entitled "The gelatinase inhibitor ACT-03 reduces gliosis in the rapid kindling model of epilepsy, and attenuates inflammation and loss of barrier integrity in vitro" where the authors aimed at assessing the novel gelatinase inhibitor ACT-18 03 as a disease-modifying drug in models of epilepsy via evaluation of neuroinflammation and by trying to test its neuroprotective effects by preserving the blood-brain barrier (BBB) integrity and reducing inflammation in vitro.

There are major comments on this work, the authors have invested in the epilepsy model and only assessed inflammation in the brain with iba1 and vimentin (which should be replaced with GFAP as a better marker for astrogliosis) and then they reverted to cell culture models using ionomycin and PMA!!! Which lowers the value of the discussed drug ACT-03 drug. Also, NeuN should be evaluated to show the lesser neural injury is occurring

Response: We thank the reviewer for the comments. In the brain we assessed microgliosis using an Iba-1 antibody and reactive astrogliosis using a vimentin antibody. We also used a GFAP antibody, but this antibody does not discriminate between resting and reactive astrocytes, leading us to conclude that the vimentin staining are more informative and we did not include the GFAP data. We would like to refer to our previous paper (doi:10.1172/JCI138332) where in the brains of the same animals NeuN has been evaluated in Supplementary Figure 15, showing a trend towards neuroprotection. For in vitro studies we focused on inflammatory factors and used PMA and ionomycin, which are regularly used in these type of studies to induce a pro-inflammatory milieu that is relevant to epilepsy. We have clarified this in the discussion by adding the following sentence “To model the environment of a pro-inflammatory state, we used PMA and ionomycin stimulation of astrocytes and endothelial cells, which has previously been shown to upregulate inflammatory mediators relevant for mimicking the changes seen during epileptogenesis 56-59”.

Point 1: First, the BBB should be evaluated in the rat model used and the ionomycin and PMA treatment for the in vitro BBB treated with ACT-30 drug represent or evidence for protection for BBB integrity.

Response 1: We have evaluated the integrity of the BBB in the rapid kindling model 9 days after the last kindling session using an albumin antibody and assessing albumin extravasation in the hippocampus. However, we did not observe remarkably extravasation of albumin in these kindled animals. We do not exclude that BBB dysfunction has occurred, but considering that animals were sacrificed more than a week after the last kindling session, the BBB might have been restored over that time.
As BBB disruption is evident in animal models of epilepsy and is observed in patients with epilepsy, we decided to investigate the effects of ACT-03 on barrier function using an in vitro model.  

We have now added the results of the albumin stainings to Supplementary Figure 1, and discussed this in the manuscript.

Point 2: The animal model should be the primary aim for the drug rather than the artificial cell culture model that relied mainly on rt-PCR which also doesn’t reflect the neuroprotection of the proposed drug

Response 2: We agree with the reviewer that the animal model is the primary aim for this drug. We have studied the effect of ACT-03’s inhibition of MMPs thoroughly in our previous study, using three different paradigms of epilepsy in an electrically induced seizure model as well as a chemically induced model of spontaneous seizure generation. To make this more clear, we have added the following text to the introduction: “Since we investigated in our previous study 29 the effects of ACT-03 on seizure activity, cognition, and neuronal loss and confirmed that MMP2 and MMP9 were inhibited using activity assays and western blot analysis, we were interested in the present study what it’s mechanism of actions are that could lead to the reduction of symptoms. Furthermore, by identifying the pathological processes that are modulated by ACT-03 treatment, this study might provide more insight in the potential uses of ACT-03 as a therapeutic target. For this purpose, we studied the effects of ACT-03 treatment on astrogliosis and micro-gliosis in the brains of kindled rats from our earlier study and further investigated the cellular and molecular alterations in vitro using astrocyte and brain endothelial cell cultures, with a focus on neuroinflammatory markers and endothelial barrier permeability.”

Point 3: The MMP-2 and 9 should be assessed for their expression and activity on the protein level rather than on the gene level which again, has lesser value being performed on the cell culture rather than on animals The same applies to the pro-inflammatory cytokines being assessed via RT-PCR and reflected minor changes.

Response 3: This was confirmed in our previous study, see also response 2.

Point 4: Based on these comments, I request that the cell culture data be repeated on the animal model to determine any value for the ACT-03 drug. I don’t think there is much scientific merit in in-vitro experiments.

Response 4: We understand the concern of the reviewer. However, the combination of results of this study showing the effect of brain inflammation and barrier function in vitro and of our previous study showing disease-modifying effect in vivo (doi:10.1172/JCI138332), suggests that the drug could be of therapeutic value for other neurological diseases as well. Thus these additional data increase the knowledge about the effect of MMP inhibition by ACT-03 on specific cell populations and pave the way to evaluation of the potential of this drug in preclinical studies as well as clinical studies.  

Minor comments:

Point 5: The authors should discuss the value of using ionomycin and PMA

Response 5: The use of PMA and ionomycin is regularly used in in vitro studies to induce a pro-inflammatory milieu that is relevant to epilepsy. We have clarified this in the discussion providing related references.

Point 6: What is meant by n=10 in the method section “Kindled rats were treated daily intraperitoneally (i.p.) with either vehicle (5.0 % 88 Tween® 80 in 0.9 % saline; n=10) or 6 mg/kg ACT-03 (n=10, Accure Therapeutics”

Response 6: Thank you for pointing this out. Initially, 10 animals per group were subjected to drug administration and kindling, however a total of 4 animals were excluded due to technical problems. This is now stated in the methods section.

Reviewer 3 Report

The authors conducted an in vivo study on rats, while the title only mentions in vitro studies. The statement "in the rapid

kindling model of epilepsy "does not directly indicate animal testing, it's not clear for everyone. The title should be changed.

In the abstract, the use of the words "(1) background", "(2) methods", "(3) results", "(4) conclusions" is superfluous, remove.

lane 69: The authors indicate a new type of inhibitor with better properties (ACT-03). What is its advantage? does it differ in structure? What is the mechanism of its action in relation to old substances? Is it selective? Has it been tested in vivo or only in vitro or in silico? It is worth explaining.

Chapter 2.1.1.

The authors present a scheme of in vivo tests, but there is no information about the "rapid kindling rat model". For the reader, this is key information to understand the entire process of the experiment and the sense of applying such a model. The authors refer to another publication that refers to the supplementary material. It is worth describing briefly and also putting a full description in the supplement.

Chapter 2.1.2.

The number of animals (n = 10, n = 10) does not agree with those listed in chapter 2.1.1 (n = 8, n = 8)

What was the volume of vehiculum?

It is not clear how the ACT-03 dose was selected.

It is not clear:

"rats were treated daily intraperitoneally (i.p.) (...) for one week, ACT-03 injection (volume of 10 mL/kg) took place one hour before kindling stimulations on days 1, 2, and 3 between 8:00 and 9 : 00 A.M. " next "Rats were decapitated one day after the re-kindling session (23 days after electrode implantation; 10 days after the last ACT-03 injection.".

Were the injections only on days 1, 2, 3, or throughout the one week? How long after the surgery did treatment start? It is worth presenting the diagram of the experiment in graphic form.

line 132, 147, 182: I am against writing "as previously described", "described elsewhere". The work itself should explain the entire work protocol and the methods used, and should not refer to other publications for details. The description does not have to be very detailed, but it should present the methods, and techniques used so that it is clear to the reader and possible to repeat.

Chapter 2.2.1

It is not clear from how many fetus cells are obtained? Were they pulled or treated separately as a different phenotype? Information on the number of donors is available only late in chapter 2.2.4.

Chapter 2.2.3

The chapter is completely unclear. Did the authors read the paper before submitting it? Figure 12 - There is no such at work. The entire chapter in small font. The cell culture fragment should be separate, chapter 2.2.3. should start with measuring the amount of RNA in the protected material, then rt-qPCR analysis as the title says.

Chapter 2.2.4., 2.2.5.

these chapters should be before chapter 2.2.3, cells methods, and next to the analysis part (2.2.3.).

Figure 4A is labeled on line 202 (page 5) while the figure appears on page 14, line 404. The figure should be moved to where it is first cited. If the figure denotes different chapters, it should be divided so that it is in the place where it is described.

Chapter 2.3.

what level of statistical significance was used?

Figure 1.

It should be divided into two figures so that you can put figures/graphs and their description on one page, thanks to which you will be able to remove the empty space on page 7. The graphs Description of the figure will be also reduced and well presented.  Figures are very illegible, it is not known what description is for which graph (top? Bottom?). Larger gaps or frames should be used.

Figure 2.

As above.

Figure 3.

As above.

Author Response

Point 1: The authors conducted an in vivo study on rats, while the title only mentions in vitro studies. The statement "in the rapid kindling model of epilepsy "does not directly indicate animal testing, it's not clear for everyone. The title should be changed.

Response 1: We have adjusted the title so that the rapid kindling model clearly refers to an in vivo study.

Point 2: In the abstract, the use of the words "(1) background", "(2) methods", "(3) results", "(4) conclusions" is superfluous, remove.

Response 2: These words are parts of the journal’s template.

Point 3: lane 69: The authors indicate a new type of inhibitor with better properties (ACT-03). What is its advantage? does it differ in structure? What is the mechanism of its action in relation to old substances? Is it selective? Has it been tested in vivo or only in vitro or in silico? It is worth explaining.

Response 3: We have now added more information about ACT-03 to the introduction based on our previous study (Bertran et al. 2020: 10.1016/j.bioorg.2019.103365)

Point 4:

Chapter 2.1.1.

The authors present a scheme of in vivo tests, but there is no information about the "rapid kindling rat model". For the reader, this is key information to understand the entire process of the experiment and the sense of applying such a model. The authors refer to another publication that refers to the supplementary material. It is worth describing briefly and also putting a full description in the supplement.

Response 4: We apologize for this and have now added a detailed description of the in vivo model in the supplementary materials.

Point 5:

Chapter 2.1.2.

The number of animals (n = 10, n = 10) does not agree with those listed in chapter 2.1.1 (n = 8, n = 8)

What was the volume of vehiculum?

It is not clear how the ACT-03 dose was selected.
It is not clear:

"rats were treated daily intraperitoneally (i.p.) (...) for one week, ACT-03 injection (volume of 10 mL/kg) took place one hour before kindling stimulations on days 1, 2, and 3 between 8:00 and 9 : 00 A.M. " next "Rats were decapitated one day after the re-kindling session (23 days after electrode implantation; 10 days after the last ACT-03 injection.".

Were the injections only on days 1, 2, 3, or throughout the one week? How long after the surgery did treatment start? It is worth presenting the diagram of the experiment in graphic form.

Line 132, 147, 182: I am against writing "as previously described", "described elsewhere". The work itself should explain the entire work protocol and the methods used, and should not refer to other publications for details. The description does not have to be very detailed, but it should present the methods, and techniques used so that it is clear to the reader and possible to repeat.

Response 5: Our apologies for this discrepancy in animal numbers. This is due to the loss of some animals. This is now added to the text. Vehicle injections had the same volume as ACT-03 injections. This has been clarified in the text. We have added a detailed description of the in vivo model to the Supplementary Materials.

Point 6:

Chapter 2.2.1

It is not clear from how many fetus cells are obtained? Were they pulled or treated separately as a different phenotype? Information on the number of donors is available only late in chapter 2.2.4.

Response 6: The number of donors used is discussed in the several method sections as well as the results section under the respective figures. For clarity, we have now added this information to all relevant method sections. Data from experimental conditions were normalized to the control condition of the respective donor allowing them to be pooled.

Point 7:

Chapter 2.2.3

The chapter is completely unclear. Did the authors read the paper before submitting it? Figure 12 - There is no such at work. The entire chapter in small font. The cell culture fragment should be separate, chapter 2.2.3. should start with measuring the amount of RNA in the protected material, then rt-qPCR analysis as the title says.

Chapter 2.2.4., 2.2.5.

these chapters should be before chapter 2.2.3, cells methods, and next to the analysis part (2.2.3.).

Response 7: Thank you for expressing your detailed concerns with the placement of these chapters. We have replaced them accordingly as well as corrected the error concerning the text ‘Figure 12’.

Point 8:

Figure 4A is labeled on line 202 (page 5) while the figure appears on page 14, line 404. The figure should be moved to where it is first cited. If the figure denotes different chapters, it should be divided so that it is in the place where it is described.

Response 8: As this concerns a figure containing results, the reference to Fig 4A has been relocated to the results section.

Point 9:

Chapter 2.3.

what level of statistical significance was used?

Response 9: A p-value < 0.05 has been used to indicate significance. This has been added to Section 2.3.

Point 10:

Figure 1.

It should be divided into two figures so that you can put figures/graphs and their description on one page, thanks to which you will be able to remove the empty space on page 7. The graphs Description of the figure will be also reduced and well presented.  Figures are very illegible, it is not known what description is for which graph (top? Bottom?). Larger gaps or frames should be used.

Figure 2.

As above.

Figure 3.

As above.

Response 10: Thank you for your concern, we will discuss the size and placement of the figures with the editor and will change the figures accordingly.

Reviewer 4 Report

You write of` `epilepsy', but your experimental model is one of focal (temporal lobe) epilepsy, as you specified in your earlier paper (Ref #29). Your interpretations may not necessarily apply to other types of seizure disorders, e.g. genetic generalised epilepsies. Perhaps you should be more specific regarding the possible applicability of your work.

In Fig 1 the ? haematoxylin stained neurone nuclei looked reduced in number in proportion to the ease of seizure genesis. Seizures probably originate in neurones and not glial cells, and if the lost neurones are inhibitory in function, the glial changes may be reactions to inhibitory neuron loss and only surrogates measures of seizure genesis potential.

Will your inhibitor cross the blood-brain barrier? If not, it may not have much potential as an antiepileptic agent. 

Author Response

Point 1: You write of` `epilepsy', but your experimental model is one of focal (temporal lobe) epilepsy, as you specified in your earlier paper (Ref #29). Your interpretations may not necessarily apply to other types of seizure disorders, e.g. genetic generalised epilepsies. Perhaps you should be more specific regarding the possible applicability of your work.

Response 1: We thank the reviewer for this response. Indeed, the model we have used mimics the pathological characteristics of focal temporal lobe epilepsy and therefore the results cannot be directly translated to other epilepsy types. We have adjusted our language when discussing epilepsy in the manuscript.
Nevertheless, the evidence of ACT-03’s effect in brain inflammation and BBB disruption might point towards a potential therapeutic effect beyond temporal lobe epilepsies, such as epilepsies induced by traumatic brain injuries, strokes, infections and even genetic epilepsies. Future studies should focus on other epilepsy types as well as more neurological disorders, to determine whether MMP inhibition through ACT-03 might be beneficial in those environments as well.

Point 2: In Fig 1 the ? haematoxylin stained neurone nuclei looked reduced in number in proportion to the ease of seizure genesis. Seizures probably originate in neurones and not glial cells, and if the lost neurones are inhibitory in function, the glial changes may be reactions to inhibitory neuron loss and only surrogates measures of seizure genesis potential.

Response 2: Thank you for raising this good point. We have examined the number of hilar neurons, in our previous paper, Suppl. Fig. 15, and there was a trend towards less neuronal loss in the ACT-03 (there referred to as IPR-179) treated animals. We would therefore hypothesize that the glial changes are not merely a result or representation of the seizures. Since it’s known that glial functional changes can influence the function of neurons (doi: 10.3389/fneur.2020.591690 and others), we found it important to further examine this. However, in the animal model it is impossible to unravel that because of the decreased seizure activity, hence we have turned to in vitro models for investigating the seizure-independent effects of ACT-03.

Point 3:

Will your inhibitor cross the blood-brain barrier? If not, it may not have much potential as an antiepileptic agent.

Response 3: Indeed, ACT-03 can cross the BBB as discussed by Bertran et al 2020 (10.1016/j.bioorg.2019.103365). We have now referred to this in the manuscript.

Round 2

Reviewer 2 Report

again, the authors were not responsive to the constructive comments and reverted always to the response: we have done this in our previous work.

when GFAP the golden standard for astrogliosis is requested and the authors considered nonrepresentative and used vimentin instead, I cant ut request that this IF be performed.

as you are assessing this current model, we need to assess the   MMP 9 needs to be performedon the animal protein level rather than rt-PCR

Author Response

Response to Reviewer 2 Comments

again, the authors were not responsive to the constructive comments and reverted always to the response: we have done this in our previous work.

when GFAP the golden standard for astrogliosis is requested and the authors considered nonrepresentative and used vimentin instead, I cant ut request that this IF be performed.

as you are assessing this current model, we need to assess the   MMP 9 needs to be performedon the animal protein level rather than rt-PCR

Response:

We are very sorry to hear that the responses were not to your satisfaction despite the addition of an extra experiment.

GFAP is indeed the most commonly used marker to detect astrocytes, and as suggested, we have performed GFAP immunohistochemistry and quantified this. However, we did not observe changes between vehicle-treated kindled rats and ACT-03-treated kindled rats, in contrast to vimentin immunohistochemistry (which is specific for reactive astrocytes and also commonly used). We have now included this information in the methods and results and added representative pictures of GFAP stainings in Supplementary Figure 1, and Supplementary Table 1.
Because of the high constitutive presence of GFAP, interpretation of GFAP immunoreactivity is troublesome, especially in a region such as the hippocampus. This is discussed by Escartin et al. 2021 (doi: 10.1038/s41593-020-00783-4) and it is even argued there that the GFAP changes “may reflect physiological adaptive plasticity rather than being simply a reactive response to pathological stimuli”. For this reason, we chose to focus on vimentin expression, which is strongly upregulated in both human and rat reactive astrocytes (Liddelow & Barres 2017 10.1016/j.immuni.2017.06.006; Gorter et al. 2002 doi: 10.1046/j.1460-9568.2002.02078)

To investigate MMP9 protein expression in the rat brain, as the reviewer requests, we have performed immunohistochemistry for this protein and quantified this. We observed that MMP9 expression in glial cells was increased in the kindled animals, compared to the non-kindled animals. ACT-03 treatment did not result in a decrease of MMP9 (see new Supplementary Table 2). However, the mechanism of action of ACT-03 is the binding of the active site of MMP9, making it unable to cleave it’s targets. The inhibition of the protein, rather than the protein level is therefore of more value for its therapeutic potential. We have shown this in the same animals in our previous paper (Broekaart et al. 2021) using protein measurements of the short proteolytic fragment of Nectin-3 over the full protein, which is used as an indicator of MMP9 activity (van der Kooij et al. 2014, doi 10.1038/ncomms5995).

Reviewer 3 Report

Point 1: The authors conducted an in vivo study on rats, while the title only mentions in vitro studies. The statement "in the rapid kindling model of epilepsy "does not directly indicate animal testing, it's not clear for everyone. The title should be changed.

Response 1: We have adjusted the title so that the rapid kindling model clearly refers to an in vivo study.

OK.

Point 2: In the abstract, the use of the words "(1) background", "(2) methods", "(3) results", "(4) conclusions" is superfluous, remove.

Response 2: These words are parts of the journal’s template.

Please read the Instructions for the authors again carefully:

  • Abstract: The abstract should be a total of about 200 words maximum. The abstract should be a single paragraph and should follow the style of structured abstracts, but without headings: 1) Background: Place the question addressed in a broad context and highlight the purpose of the study; 2) Methods: Describe briefly the main methods or treatments applied. Include any relevant preregistration numbers, and species and strains of any animals used. 3) Results: Summarize the article's main findings; and 4) Conclusion: Indicate the main conclusions or interpretations. The abstract should be an objective representation of the article: it must not contain results which are not presented and substantiated in the main text and should not exaggerate the main conclusions.

Point 3: lane 69: The authors indicate a new type of inhibitor with better properties (ACT-03). What is its advantage? does it differ in structure? What is the mechanism of its action in relation to old substances? Is it selective? Has it been tested in vivo or only in vitro or in silico? It is worth explaining.

Response 3: We have now added more information about ACT-03 to the introduction based on our previous study (Bertran et al. 2020: 10.1016/j.bioorg.2019.103365)

OK

Point 4:

Chapter 2.1.1.

The authors present a scheme of in vivo tests, but there is no information about the "rapid kindling rat model". For the reader, this is key information to understand the entire process of the experiment and the sense of applying such a model. The authors refer to another publication that refers to the supplementary material. It is worth describing briefly and also putting a full description in the supplement.

Response 4: We apologize for this and have now added a detailed description of the in vivo model in the supplementary materials.

- Great, but remove please word: „Supplementary” from „Supplementary Figure 1.”

- next, the Figure on page 11 should be numbered Figure 2, and so on.

- The title of figure 1:  „Albumin extravasation could not be detected 9 days after the last kindling 111 session” is confusing ???

- I propose to split Figure to two separated figures: figure 1. rapid kindling rat model” and „figure 2. Albumin immunohistochemistry” – those two figures have nothing in common to be together.

The title of the Figure should be below Figure.

Point 5:

Chapter 2.1.2.

The number of animals (n = 10, n = 10) does not agree with those listed in chapter 2.1.1 (n = 8, n = 8)

What was the volume of vehiculum?

OK

It is not clear how the ACT-03 dose was selected.

The dose 6 mg/kg of ACT-03 is not explained. In the introduction other experiments with ACT-03 are cited, but there is no information about used doses and dose selection process.

It is not clear:

"rats were treated daily intraperitoneally (i.p.) (...) for one week, ACT-03 injection (volume of 10 mL/kg) took place one hour before kindling stimulations on days 1, 2, and 3 between 8:00 and 9 : 00 A.M. " next "Rats were decapitated one day after the re-kindling session (23 days after electrode implantation; 10 days after the last ACT-03 injection.".

Were the injections only on days 1, 2, 3, or throughout the one week? How long after the surgery did treatment start? It is worth presenting the diagram of the experiment in graphic form.

Still not clear. In the text you have: „daily … for one week”, next „Injections (volume of 10 mL/kg) took place one hour before kindling stimulations on day 1, 2 and 3 between 8:00 and 9:00 A.M” while on figure 1. you have 7 dots meaning 7 injections.

Line 132, 147, 182: I am against writing "as previously described", "described elsewhere". The work itself should explain the entire work protocol and the methods used, and should not refer to other publications for details. The description does not have to be very detailed, but it should present the methods, and techniques used so that it is clear to the reader and possible to repeat.

Response 5: Our apologies for this discrepancy in animal numbers. This is due to the loss of some animals. This is now added to the text. Vehicle injections had the same volume as ACT-03 injections. This has been clarified in the text. We have added a detailed description of the in vivo model to the Supplementary Materials.

OK

Point 6:

Chapter 2.2.1

It is not clear from how many fetus cells are obtained? Were they pulled or treated separately as a different phenotype? Information on the number of donors is available only late in chapter 2.2.4.

Response 6: The number of donors used is discussed in the several method sections as well as the results section under the respective figures. For clarity, we have now added this information to all relevant method sections. Data from experimental conditions were normalized to the control condition of the respective donor allowing them to be pooled.

OK.

Point 7:

Chapter 2.2.3

The chapter is completely unclear. Did the authors read the paper before submitting it? Figure 12 - There is no such at work. The entire chapter in small font. The cell culture fragment should be separate, chapter 2.2.3. should start with measuring the amount of RNA in the protected material, then rt-qPCR analysis as the title says.

Chapter 2.2.4., 2.2.5.

these chapters should be before chapter 2.2.3, cells methods, and next to the analysis part (2.2.3.).

Response 7: Thank you for expressing your detailed concerns with the placement of these chapters. We have replaced them accordingly as well as corrected the error concerning the text ‘Figure 12’.

OK

Point 8:

Figure 4A is labeled on line 202 (page 5) while the Figure appears on page 14, line 404. The Figure should be moved to where it is first cited. If the Figure denotes different chapters, it should be divided so that it is in the place where it is described.

Response 8: As this concerns a figure containing results, the reference to Fig 4A has been relocated to the results section.

OK

Point 9:

Chapter 2.3.

what level of statistical significance was used?

Response 9: A p-value < 0.05 has been used to indicate significance. This has been added to Section 2.3.

OK

Point 10:

Figure 1.

It should be divided into two figures so that you can put figures/graphs and their description on one page, thanks to which you will be able to remove the empty space on page 7. The graphs Description of the Figure will be also reduced and well presented. Figures are very illegible, it is not known what description is for which graph (top? Bottom?). Larger gaps or frames should be used.

Numer change to Figure 2, and so on.

Please consider to split this Figure to two separate figures (A-F+G-J, and K-P +Q-T) . The description could be shorter and placed directly under Figure on the same page.

Figure 2.

As above.

Please check figure 2. You have some mess on pages 13-14.

Please consider to split this Figure to two separate figures (A-F and G-N) . The description could be shorter and placed directly under Figure on the same page.

Renumber figures.

Figure 3.

As above.

Please consider to split this Figure to two separate figures (A-F and G-N) . The description could be shorter and placed directly under Figure on the same page.

Renumber figures.

Response 10: Thank you for your concern, we will discuss the size and placement of the figures with the editor and will change the figures accordingly.

OK

Figure 4.

Title of the Figure should be below the Figure it is useless to use the same description twice.

If you provide some Supplementary data (Supplementary Methods), it should be placed in Supplementary materials. Another way you should place this data in a respective chapter (M&M). For me, this data are quite important for experiment workflow, so I will propose to move it and combine to a specific chapter.

Author Response

Response to Reviewer 3 Comments

Please read the Instructions for the authors again carefully:

  • Abstract: The abstract should be a total of about 200 words maximum. The abstract should be a single paragraph and should follow the style of structured abstracts, but without headings: 1) Background: Place the question addressed in a broad context and highlight the purpose of the study; 2) Methods: Describe briefly the main methods or treatments applied. Include any relevant preregistration numbers, and species and strains of any animals used. 3) Results: Summarize the article's main findings; and 4) Conclusion: Indicate the main conclusions or interpretations. The abstract should be an objective representation of the article: it must not contain results which are not presented and substantiated in the main text and should not exaggerate the main conclusions.

Response:

Our apologies, this has now been corrected. 

Chapter 2.1.1.

The authors present a scheme of in vivo tests, but there is no information about the "rapid kindling rat model". For the reader, this is key information to understand the entire process of the experiment and the sense of applying such a model. The authors refer to another publication that refers to the supplementary material. It is worth describing briefly and also putting a full description in the supplement.

Response 4: We apologize for this and have now added a detailed description of the in vivo model in the supplementary materials.

- Great, but remove please word: „Supplementary” from „Supplementary Figure 1.”

- next, the Figure on page 11 should be numbered Figure 2, and so on.

- The title of figure 1:  „Albumin extravasation could not be detected 9 days after the last kindling 111 session” is confusing ???

- I propose to split Figure to two separated figures: figure 1. rapid kindling rat model” and „figure 2. Albumin immunohistochemistry” – those two figures have nothing in common to be together.

The title of the Figure should be below Figure.

Response: We have separated the in vivo time line and changed the figure order accordingly.

Upon request of the other reviewer, we have added a GFAP staining to the supplementary materials and combined it with the albumin staining in Supplementary figure 1 and changed the title to fit this new figure.

Point 5:

Chapter 2.1.2.

The number of animals (n = 10, n = 10) does not agree with those listed in chapter 2.1.1 (n = 8, n = 8)

The dose 6 mg/kg of ACT-03 is not explained. In the introduction other experiments with ACT-03 are cited, but there is no information about used doses and dose selection process.

Response: Selection of dosage is now explained in 2.1.2.

Still not clear. In the text you have: „daily … for one week”, next „Injections (volume of 10 mL/kg) took place one hour before kindling stimulations on day 1, 2 and 3 between 8:00 and 9:00 A.M” while on figure 1. you have 7 dots meaning 7 injections.

Response: Animals were treated with ACT-03 once daily, for 7 subsequent days.
On the first three days of this treatment, animals were subjected to kindling stimulations.
These stimulations started 1 hour after the injection of ACT-03.

Point 10:

Figure 1.

It should be divided into two figures so that you can put figures/graphs and their description on one page, thanks to which you will be able to remove the empty space on page 7. The graphs Description of the Figure will be also reduced and well presented. Figures are very illegible, it is not known what description is for which graph (top? Bottom?). Larger gaps or frames should be used.

Numer change to Figure 2, and so on.

Please consider to split this Figure to two separate figures (A-F+G-J, and K-P +Q-T) . The description could be shorter and placed directly under Figure on the same page.

Figure 2.

As above.

Please check figure 2. You have some mess on pages 13-14.

Please consider to split this Figure to two separate figures (A-F and G-N) . The description could be shorter and placed directly under Figure on the same page.

Renumber figures.

Figure 3.

As above.

Please consider to split this Figure to two separate figures (A-F and G-N) . The description could be shorter and placed directly under Figure on the same page.

Renumber figures.

Response: The original figures 1 to 3 have been split in two and are now figures 2 to 7.

Figure 4.

Title of the Figure should be below the Figure it is useless to use the same description twice.

 Response: OK

If you provide some Supplementary data (Supplementary Methods), it should be placed in Supplementary materials. Another way you should place this data in a respective chapter (M&M). For me, this data are quite important for experiment workflow, so I will propose to move it and combine to a specific chapter.

Response: Since the present study uses the same animals as our previous publication, and the behavioral seizures are not the main outcome measure for this manuscript, we have chosen to keep Supplementary methods in the Supplementary materials.

Round 3

Reviewer 2 Report

accept